# A View of Champā Sites in Phú Yên Province, Vietnam: Toward a *Longue Durée* of Socio-Religious Context

Van Son Quang [1] and William B. Noseworthy [2,*]

1   Institute of Cultural Heritage and Development Studies, Van Lang University, 69/68 Dang Thuy Tram, Ward 13, Binh Thanh District, Ho Chi Minh City 70000, Vietnam; son.qv@vlu.edu.vn
2   Southeast Asian Materials Specialist, Cornell University, Ithaca, NY 14850, USA
*   Correspondence: professornoseworthy@gmail.com

**Abstract:** Champā sites in Phú Yên province, Vietnam, were in what historians have typically called the polity of 'Kauṭhāra'. Among these, the Hồ Citadel was mentioned in recent studies of Champā citadels and Champā archaeology, but the region of 'Kauṭhāra' has yet to be analyzed with a vision for the *Longue durée* of cultural history. Drawing on the study of maps, historical documents, and archaeological evidence, we provide a more coherent understanding of Hồ Citadel in the socio-religious context of Champā, including the incorporation of Champā's Hindu–Buddhist polities of Kauṭhāra and Vijaya into what is now Vietnam; historical evidence that suggests follow up archaeological research could fruitfully focus on the Early Modern period of history. Our findings suggest the region was one of the longest-occupied Champā regions, despite a comparative lack of focus on archaeological studies in the area versus Champā sites further northward. Future archaeological work should not only focus on the very earliest finds but rather the *Longue durée* of persistent settlement patterns. Thus, we hope to inspire continued and more direct collaborations between historians and archaeologists for the benefit of advancing research in the study of local and transregional understandings of Asia in the Humanities and Social Sciences.

**Keywords:** central Vietnam; Champā; Hindu–Buddhist cultures; historical archaeology





## 1. Introduction

The Hindu–Buddhist Champā civilization was located along the coast of central Vietnam from the Hải Vân pass to Mui Kê Ga and into the hinterlands of the foothills of the Annamite Chain. Champā lasted in various forms from the 2nd century CE until 1832 CE when the last polity (Pāṇḍuraṅga) was annexed by the Nguyễn Vietnamese Empire. Vickery (2011, p. 364) divides Champā into four riverine regions: (1) now in Quảng Nam province and northward; (2) now in Bình Định province; (3) now in Khánh Hòa province; and (4) now in Ninh Thuận and Bình Thuận provinces. Archaeologists have mostly focused their work in the first two regions, although some studies mention our region, which is between regions (2) and (3) the in Phú Yên province (Southworth 2011; Đỗ et al. 2017; Lâm 2019; Barocco et al. 2019). Vickery (2011, p. 365) states that this location, the Đà Rằng River valley[1], is one of two important riverine settlement regions that has not received adequate attention concerning the study of the history of Champā[2]. Vickery (2011, p. 365) highlights a 5th century Sanskrit inscription, found at the mouth of the Đà Rằng River, and points to the 'later' Hồ Citadel[3]—about 15 km inland—as two of the important sites in the region. However, more recent archaeological studies have suggested that there is evidence of a settlement at the location became Hồ Citadel, and this could be as early as the 2nd century, suggesting that this very early period of settlement may parallel the Lín Yì settlements developments further north (Lâm 2019). Recent historical analyses show the settlements in the region only came under Vietnamese control in the 17th century, despite Vietnamese claims to have completely conquered the area in the 15th century (Ken 2011;

Noseworthy 2015). Through a *longue durée* vision of this region, we not only connect the vicinity of the Hồ Citadel to transregional networks that impacted the history of Hindu-Buddhist Champā but also trace the history into the deeply contested Early Modern Period, suggesting that future research focusing archeological digs on the Early Modern sites could be particularly fruitful. The arguably most influential use of the Braudelian concepts in Southeast Asian history to date has been by historian of Southeast Asia, Barbara Andaya (2016), who drew on the concept of the Mediterranean and "Ocean Studies" to begin to examine the littorals of Southeast Asia, such as the South China Sea and cultural history. Noseworthy (2014) then drew on the concept of the *longue durée* specifically to discuss the South China Sea as a space of transregional cultural flows. While we build on the foundations of these earlier studies, we are interested in how the *longue durée* helps us consider trajectories for studies at the intersection of history and archaeology.

To inform our analyses, we have collected Chinese, Vietnamese, Cham, and European historical documents, along with the extensive scholarly assessment of archaeological fieldwork and the surveys conducted by our first author, an archaeologist himself, to complete this study. We begin with (1) an assessment of existing archaeological work that has been completed concerning the Hồ Citadel, followed by (2) a revised vision of the Champā sites in Phú Yên within the context of Kauṭhāra and Vijaya, Champā, before concluding with (3) a historical assessment of the 'end of the days' of the Hồ Citadel according to historical sources. This analysis allows us to establish proper research trajectories, in the conclusion, for future combined historical and archaeological research. The evidence suggests future studies should rely on interdisciplinary teams of historians and archaeologists to make combined assessments of the *Longue durée* of local settlements with a mind for transregional connections of religious and trade networks, along with historical records. These future analyses could focus on the neglected Early Modern epoch of history in greater detail, providing us with a better sense of how locals in this region were incorporated into and strategically resisted incorporation into Vietnamese rule.

## 2. Archaeological Studies of Champā Sites in Phú Yên

In the late 19th and early 20th centuries, French scholars began archaeological surveys of Champā civilization sites. In 1909, Henri Parmentier released an initial survey of the Hồ Citadel, providing descriptions and detailed drawings of the sites (see Figure 1).

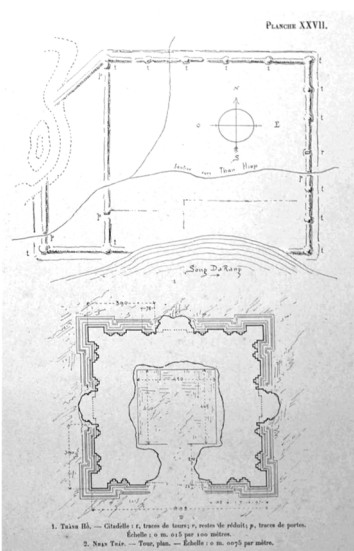

**Figure 1.** Hồ Citadel (**top**) and Nhạn Tower (**bottom**) by Parmentier (1909b, Planche XXVII).

According to Parmentier (1909a, pp. 137–38), the citadel was on the 'left' (north) bank of the Đà Rằng River, 15 km from where the river delta flows into the South China Sea. The sides were generally square, at 600 m each. There was also a second section (west) that is

square on the south side and triangular on the north side, with part of the embankment relying upon the nearby hill as a natural barrier. The Hồ Citadel also had evidence of a 30 m wide moat to protect the north and east walls. There was evidence of watchtowers on all sides, except the side protected by the nearby mountains, with six along the north wall and seven on the east wall, inclusive of the corner towers. Yet the southern wall had already collapsed by the early 20th century, likely resulting from the flows of the Đà Rằng River, but it still had two mounds in the west corner. The east gate had an opening that suggested water might have flown through it, as did the north and both of the western walls. Today, our first author found that the walls are often barely visible, but they are notable as they appear as changes in the landscape (see Figure 2).

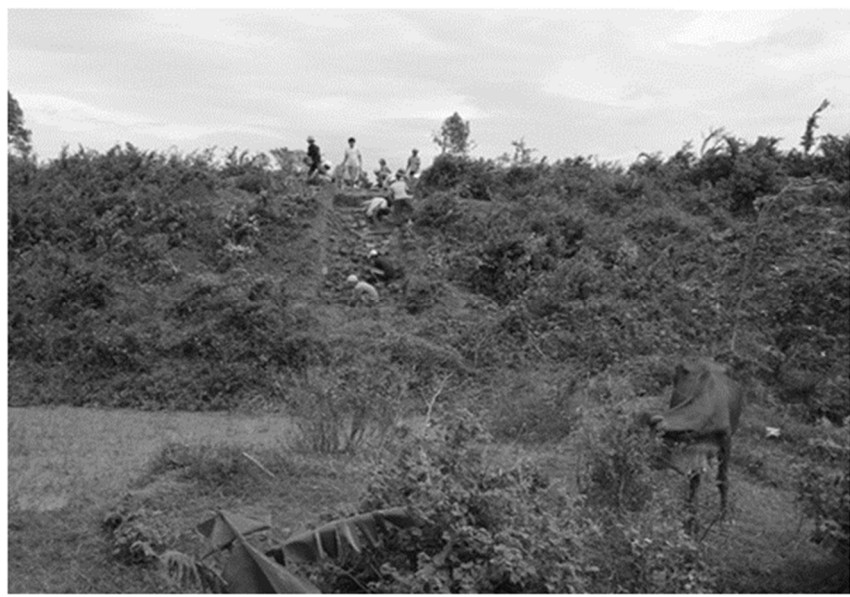

**Figure 2.** Edge of Hồ Citadel, Blended into Edge of Agricultural Landscape and the Local Forest (Photograph by Văn Sơn Quảng).

As we can see from Figure 2, the edges of the walls of the citadel blend very naturally into the local agricultural landscape, which has included farmland for Vietnamese families living in the area. Nonetheless, the brick constructions of the Hồ Citadel were very large by Champā standards, rising to approximately 0.1 m, with deep red hues that are sometimes crimson, almost purplish, in color. The production site appears to have been located relatively nearby at Phước Thịnh, along the north–south axis of the region and across the Đà Rằng River, based on finds of numerous broken bricks at that location (Parmentier 1909a, pp. 137–38; Ngô 2002, pp. 2, 10–11). Vietnamese nationals have surveyed Champā relics in Phú Yên many times since Parmentier's works, especially in the 1990s and early decades of the 21st century. A substantial expedition led by Lê Đình Phụng, Vietnam's Institute of Archaeology, and the Phú Yên Museum was completed in 2003; on the other hand, a later survey in 2008–2009 was conducted by a team affiliated with the University of Social Sciences and Humanities. These surveys emphasized the importance of the Hồ Citadel as a military, political, and cultural center (Ngô 2002; Nguyễn 2004, 2010; Đặng et al. 2009; Lâm 2019). However, most of these examinations focused on specific periods of Champā history, especially the very earliest periods of occupation. There was little interest in conducting historical or archaeological surveys of this region covering the early modern period of history. They essentially took for granted a singular narrative of Vietnamese conquest and did not investigate the potentialities of the contested nature of this region during the Early Modern historical epoch.

There are three important hypotheses regarding the dating of settlements in this area. The first is based on deep analysis of roof tiles, *kendi*, and other materials dating the site to as early as the 2nd century, with more materials dating to the 4th to 5th century. However,

the ramparts in the area likely date mostly to the 5th through 7th century in their earliest forms of construction. A second hypothesis was based primarily on materials found in the 2003 and 2008–2009 archaeological surveys, including the Củng Sơn site and the Chợ Dinh rock inscription, which argues that the Hồ Citadel and surround sites mostly date to the 5th to 7th century in terms of their earliest constructions. Finally, based on an analysis of the larger constructions of the surrounding Nhan and Yang Praong towers, there is strong evidence of the occupation of this region during the 11th to 13th centuries, during which time the earlier polity of Kauṭhāra was almost certainly subordinate to the cultural and political center of Vijaya. Together, these finds point clearly toward a deep intertwining of Buddhism and Hinduism in the historical past of this locality as what scholars tend to refer to as Hindu–Buddhism (Lê and Phạm 2004; Lâm and Nguyễn 2009, pp. 45–62; Lâm 2019). Barocco et al. (2019) use the evidence from the second hypothesis to suggest this locality was a competitor of Champā settlements in the Thư Bồn River valley during the early period of Champā history. Indeed, the local preservation work, partnered with the Phú Yên Museum, and devoted to the development of the location as a stop for tours in the local provinces' tourist industry has highlighted the same period. The National Historic Marker for the site, established in a 2005 decision, indeed highlights the 5th to 7th century construction hypothesis. As a consequence, neither archaeological digs nor broader conceptions of history have focused on the later periods of the history of the polity of Kauṭhāra, especially from the 15th to 17th centuries when the region was much more contested. While it is true that military conflicts certainly destroy archaeological remains, they also leave their own archaeological remains. Furthermore, we know from historical sources that Islam was gaining prominence among Champā populations further southward during those very transitionary centuries. How was Kauṭhāra related to the other Champā polities, especially the neighboring polities of Vijaya and Pāṇḍuraṅga throughout the centuries in question? This is the question to which we shall now turn.

### 3. Phú Yên Sites and Champā *Negara*

Historians of Champā typically describe the civilization as comprised as five key Hindu–Buddhist "*Negara*"[4] polities that rose and fell over time: Amarāvatī, Indrapura, Vijaya, Kauṭhāra, and Pāṇḍuraṅga. Although this vision differs from the earlier Orientalist portrayal of a single kingdom of Champā, it became the dominant scholarly understanding of Champā by the late 1980s and one that pervades through quite recent scholarship, with some notable exceptions and potential modifications. In most contemporary scholarly understandings of Champā history, the polity of Kauṭhāra typically is centered on what is now Nha Trang[5] but includes areas that are now associated with Khánh Hòa and Phú Yên provinces (Po 1988, 1989; Ken 2011; Lafont 2011; Weber 2014, 2019). We know from studies of Champā epigraphic records that Kauṭhāra became prominent in the 8th century, had troubles in the 9th century, and was restored in the 10th century (Lafont 2011). Then, by the 12th century, Vijaya, to the north of Kauṭhāra in Bình Định province, emerged as a prominent center (Đỗ et al. 2017).

The reason why the above timeline is important is that the only notable Champā tower nearby the Hồ Citadel is the 12th century construction of the Nhạn Tower[6]. Thus, the Nhạn Tower itself may well correspond with the influence of Vijaya in the region of what is now Phú Yên. Yang Praong, which is somewhat of an anomaly in that it is far off in the uplands, is thought to be a 13th century construction and seems to be evidence of the far-reaching order of the Vijaya polity. Vijaya appears to have reigned as the most powerful polity in the region until the brutal attacks on Champā at Vijaya in the 15th century. During these campaigns, the Vietnamese army of Lê Thánh Tông supposedly went as far south as Thạch Bi—now Núi Đá Bia—before deeming the surrounding area conquered and returning northward. Yet historians of Champā and Cham Studies tend to argue for the surrounding area in what is now Phú Yên almost immediately returning to the hands of the last two polities: Kauṭhāra and Pāṇḍuraṅga. In the view of most historians of Champā Studies, the vicinity of Kauṭhāra remains subordinate to Pāṇḍuraṅga; that is, until the area was

subsumed by another Vietnamese invasion in 1611, annexing much of what is now Phú Yên, and still another invasion in 1651–1653, resulting in the annexation of the center of Kauṭhāra at Aia Trang—now Nha Trang, Khánh Hòa province (Po 1988, 1989; Lafont 2011; Weber 2014, 2019). In this vision, Kauṭhāra is relatively cohesive, stretching from the decisively Hindu site of Bimong Po Inâ Nâgar[7] at Aia Trang on the Cái River to areas just north of Mùi Đại Lành (Cape Varella), which is just south of what is now Tuy Hòa, Phú Yên province. Yet, note the locations of the Nhạn Tower and several archaeological sites in Phú Yên proposed as important for Champā history according to Barocco et al. (2019), along with their relative proximity to Vijaya and associated sites, as well as Yang Praong, and relative distance to Kauṭhāra and sites in Pāṇḍuraṅga, further south (Figure 3). Yang Praong, the center of Kauṭhāra, and the center of Vijaya are virtually equidistant from one another, with the Đà Rằng River Valley and Nhạn Tower virtually as close to Vijaya as they are to Kauṭhāra.

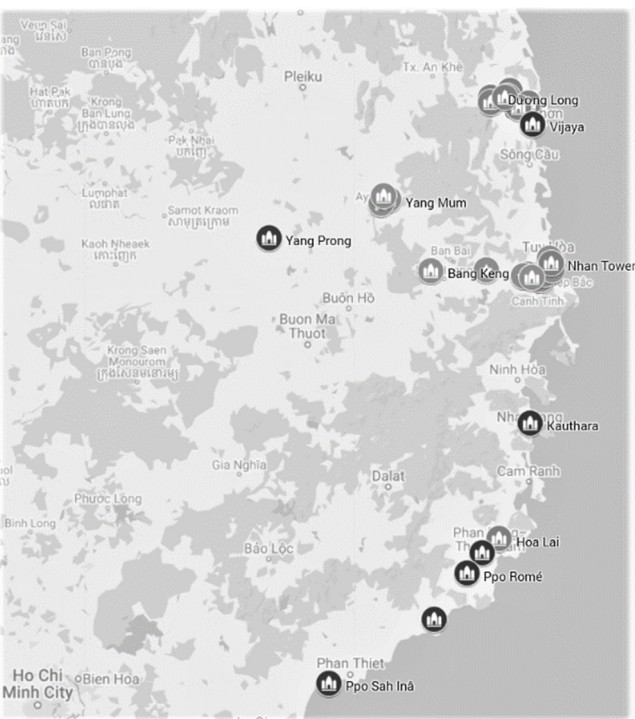

**Figure 3.** Champā Archaeological Sites: Vijaya, Kauṭhāra, and Pāṇḍuraṅga (Designed in Google Maps by Noseworthy).

*3.1. Champā Relics and Archaeological Sites in Phú Yên Province*

There are still many archaeological remains in Phú Yên province that are critical to study for the transition of the settlements in the area from Champā control to Vietnamese control. In addition to the Hồ Citadel, we have the Hồ Sơn pagoda, the 12th century Nhạn Tower, the 12th–14th century Núi Bà Tower site, the Phú Lâm site, the Tuy Hòa Buddhist reliefs, the 5th century Chợ Dinh inscription of Bhadravarman [C. 41], and the 5th century terracotta stone figures of Củng Sơn (Lafont 2011; Schweyer 2012, pp. 106, 110). Other archaeological sites in the province include Cẩm Thạch, Đồng Miếu site, Mỹ Lệ, Phú Hòa, Phước Lộc, Núi Chóp Chài, and Bảo Tịnh pagoda according to Barocco et al. (2019). The Chợ Dinh inscription was found right at the base of the Nhạn Tower, while Củng Sơn is upriver and quite far away. In between the two, Đồng Miếu is an important newly found Champā site that seems to date to the same period, being the 4th to 5th century. However, Chùa Bảo Tịnh and Chùa Hồ Sơn are currently Vietnamese Buddhist sites, with temple construction dating to the 17th and early 18th century, respectively. In other words, in the case of these sites, older Champā sites were incorporated into Vietnamese Buddhist practices by the Early Modern period. As we can see in Figures 3 and 4, many of these

sites were located quite far from the coast and upstream, although the sites more clearly incorporated into Early Modern Vietnamese Buddhism, which remain extant, are closer to the delta region. Still, we should also keep in mind that we have evidence that the flow of the Đà Rằng River has changed since the time of the Parmentier. Therefore, we can suspect that, if the flow of the river has varied in just one century, it probably changed courses several times in previous centuries, and in a significant fashion, from the very earliest period of settlement hypothesized by Lâm (2019) in the 2nd century, throughout the period of Vietnamese conquest and settlement of this region by the 17th century. What this means is that we should be cautious about drawing conclusions around the locations of the Bảo Tịnh pagoda and Hồ Sơn pagoda by terms of their proximity to the river and the coast. Nonetheless, based on patterns of conquest elsewhere, it is at least within the realm of possibility that the more downstream locations were more solidly in Vietnamese control at earlier historical periods, whereas upstream locations remained more contested.

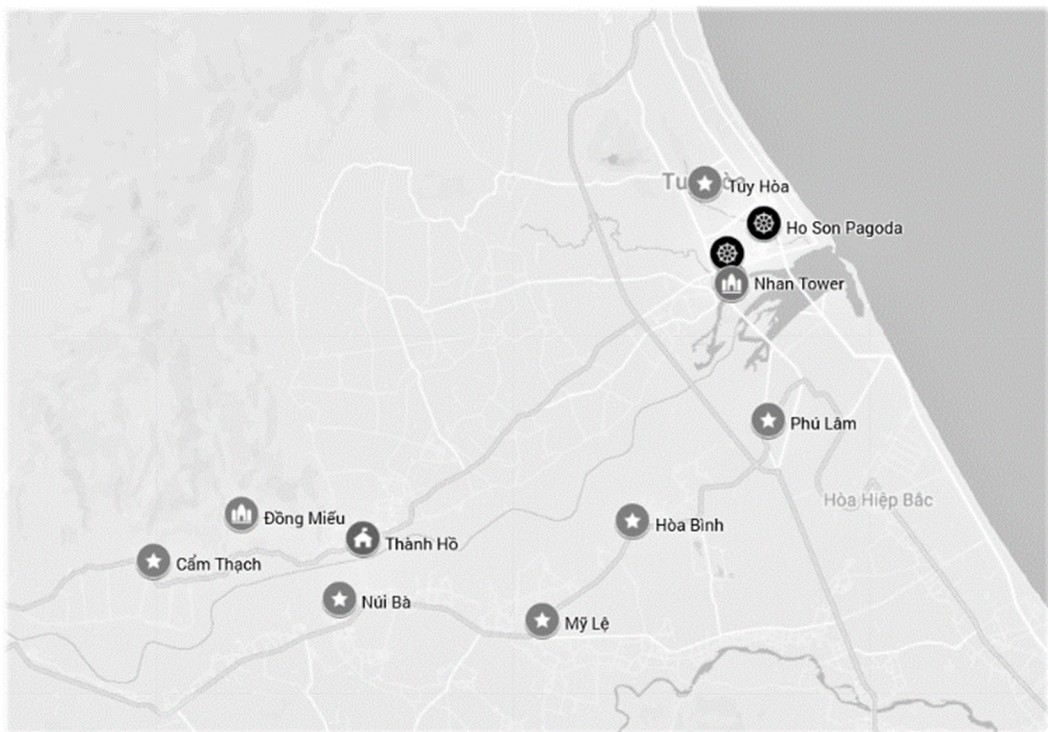

**Figure 4.** A collection of key archaeological sites in Phú Yên (Designed in Google Maps Noseworthy).

*3.2. Relations between Hồ Citadel and Champā Relics in Phú Yên Described in Parmentier's Account*

Most archaeological assessments of Phú Yên in the recent publication have not included a reflection of finds described in Parmentier's work. In part, this is because several pieces described in Parmentier's inventory have since been lost. In addition to the Nhạn Tower and the Hồ Citadel, Parmentier described one other significant archaeological site: the Núi Bà site. Parmentier described Núi Bà as opposite to the Hồ Lake, across the Đà Rằng River, and located near a dense mound of trees at 50–60 m in elevation. He also described a 'Bà Pagoda' in Phước Tịnh village, Hòa Bình canton, Tuy Hòa district. Further, he also described a Champā temple on top of the mountain, with the remaining architecture having a mess of bricks, decorative stone plates, statues from a temple, and other evidence, concentrated at the An Nam Pagoda, which was also built from Champā bricks (Parmentier 1909a, 1918; Ngô 2011, pp. 282–86). Parmentier proceeds to describe six other objects. The first is a terracotta slab (0.04 m thick, 0.28 m tall, and 0.21 m wide) that is fragmented with major parts intact. On the front side, a Buddha image sits on a lotus in front of a canopy created by a *nagā* serpent. On either side of the Buddha image, there were two slender temples resting upon zoomorphic figures. Anthropomorphic figures were cross-legged,

with their hands in their lap and a bump on the top of their heads. There was also an inscription in the Champā script on the back (Parmentier 1909a, p. 134).

Because there is no jewelry on these figures—or other tell-tale signs of Brahmins, such as Brahmanical cords that are common in Champā art—mentioned in the description, we can reason these figures are not Brahmins. We also know from the field of Buddhist studies that the *nagā* serpent mentioned must be Mucalinda, who emerged from the tree the Buddha meditated under to wrap seven times around the body of the Buddha and cover the ascetic with its body to protect him from insects and the elements while he meditated under the Bodhi tree. The additions on the tops of the heads of the cross-legged figures are likely *uṣṇīṣa*, which are the 32nd of the 32 great signs of the Buddha. The second object was a lushly carved lion head, featuring large fangs and slightly crescent or horn-shaped eyes (0.47 m tall, 0.75 m wide, and 0.13 m thick) (Parmentier 1909a, p. 134). This *siṃha* image is relatively common in Champā art and is associated with temple entrances. The third was an anthropomorphic figure attached to large steel (0.4 m wide, 1.3 m tall, and 0.25 m thick) with decorations adorning the backside. The figure stood upright, wore a cloth skirt (*sampot*), which is a brazier-like horizontal band that covered the chest of the statue, and a headdress to contain the hair (*mukuṭa*) (Parmentier 1909a, pp. 134–35). Next, Parmentier describes an image of what he believed to be Lakshmi, as the image was a slender and long body, with raised breasts and arms in front of the body holding lotus buds, with the hands raised to about shoulder height. He also described the image as holding a small cage in one hand and an empty plate in the other. She wore jewelry, including earrings, and also wore a *mukuṭa* as well as a sarong dress. The four-armed goddess was seated cross-legged on a lotus, and the entire object was a large leaf-shaped stone (0.65 m wide, 0.85 m tall, and 0.23 m thick) (Parmentier 1909a, p. 135). Such objects are commonly thought to be stone covers for wooden doors of Champā temples.

The Lakshmi image is significant when also considering the fifth image Parmentier describes, a chiseled image on a stone tablet (0.4 m wide, 0.85 m tall, and 0.24 thick), with an inscription. The image is a representation of Śiva with a Nandin bull, with the Śiva image holding a trident and its head adorned by a characteristic third eye. The anthropomorphic Śiva wears Brahman cords, a double-tiered *mukuṭa*, a piece of cloth around the chest, and a *sampot*. While the inscription was on the back of the slab, there were also three holes in the center, likely used to affix it physically to a temple (Parmentier 1909a, pp. 135–36). Typically, in South Asian understanding, Lakshmi is the feminized divine power who provides the possibility of the exteriorization of Vishnu, and she is often portrayed as his consort. Yet, the presence of Śiva does not negate that this figure could be one of many representations of Lakshmi in Champā art, as Lakshmi is often portrayed alone. The last major figure Parmentier described, however, had a direct connection to Śiva: his son Gaṇeśa. In this portrayal, Gaṇeśa was seated cross-legged, with his right hand on his knee. The left hand was broken, so it was not possible to see whether his hand was holding a bowl or touching his head, as it often does. The five-tiered *mukuṭa* is notable, although the only jewelry that was described was five pearl beads on bracelets on the upper arm (Parmentier 1909a, pp. 136–37). Lakshmi, Śiva, and Gaṇeśa indicate a type of classical religious form that was prominent in Champā art from the 7th through the 14th century, as well as a transregional religious context of Southern Asia (South and Southeast Asia).

In addition to the aforementioned more substantial objects described in Parmentier's account, there was also a series of smaller objects that were found. These include a figure on a leaf-shaped stone (0.9 m wide, 1.2 m high) with a two-tiered mukuṭa, a yoni basin decorated with lotus flowers, a lotus-shaped pedestal (1 m wide, 0.35 m tall), a flagstone slab, a Makara shaped cornerstone, and a nandin cow image. The inscriptions on the Buddha image include an invocation of the *Ye dharmā* formula and lines referencing the 6th century of the *śaka* calendar. The major inscription on the back of the aforementioned third major object included 14 lines written in the Cham language (Parmentier 1909a, pp. 136–37). The evidence from Parmentier's account of Phước Tịnh village, Hòa Bình canton, Tuy Hòa

district, and 'Bà Pagoda' caused later archaeologists to revisit this site in hopes of finding more evidence to illuminate the history of Champā culture in what is now Phú Yên.

### 3.3. Núi Bà and Champā Styles

Staff from the Phú Yên Museum and Vietnam's Institute of Archaeology conducted a series of excavations in the 1990s, especially at Núi Bà. These excavations revealed the foundation of a tower, 8.6 m on each side, with walls that are 2.3 m thick. The bricks were bright red to pale yellow in hue and quite hard, ranging from 35 cm × 15 cm × 40 cm to 40 cm × 19 cm × 8 cm in size. They also found a decorative cornerstone from the top of the tower body with a Makara (#: 90NB: 06; 1.15 m long, 0.85 m wide at the widest point, 0.45 m wide at the narrowest, and 0.15 m thick) and pieces from a *linga-yoni* structure (#: 90NB: 11). They also found a stone relief of Garuda. The Phú Yên Museum then collected the Champā stone artifacts and moved them into storage for the sake of 'preservation' (Lê and Nguyễn 1992, pp. 54–61; Đặng et al. 2009). The Museum placed these objects with the other Champā pieces they already had in their collections, including four stone square pillars (from 0.25 m to 0.35 m wide and from 2.7 m to 3 m long), a stone door (2.7 m long, 0.37 m wide, and 0.27 m thick), ten stone spiers with pyramid decorations on each level of the spiers (from 0.33 m to 0.4 m in height)—shaped with the bottom being a square-based and the upper cylinder protruding from two layers of lotus petals—as well as bow-shaped stones carved with leaf patterns (0.8 m long, 0.14 m wide, and 0.1 m thick), and 22 decorative shapes from the corners of stone temples and terracotta constructions (Đặng et al. 2009; Ngô 2011, pp. 286–88). The terracotta Buddha image, the standing statue attached to the stele with the inscription, the image of Lakshmi, the image of Śiva and the *nandin*, and the Gaṇeśa figure, mentioned by Parmentier, have all since been lost. Nevertheless, we can combine the evidence with the later Núi Bà excavations to suggest that this region was an important part of Champā's Hindu-Buddhist religious culture.

The foundations of the Núi Bà site suggest a tower over 20 m in height, which is roughly equivalent to the size of the Hindu Ppo Klaong Garai tower of Pāṇḍuraṅga and now just outside Phan Rang, Ninh Thuận province (2.14 m thick walls and 8.2 m long at their edges, with a 22.3 m tall tower). However, because the tower is no longer present at this site, dating the potential era of construction is quite tricky. Scholars have suggested that one of the cornerstones visibly resembles the 'Phoenix bird' and locals refer to cornerstones they have found at Núi Bà as 'Phoenix Tails'. Upon examination, these are solid decorative blocks with sharp tips that arch forward similarly to hooks. The 'tail' of these pieces is embellished by small pitches protruding back from the arches. These styles closely resemble the cornerstones of the Thủ Thiên and Cánh Tiên towers of Bình Định province (Vijaya) and the Ppo Klaong Garai temple of Ninh Thuận province (Pāṇḍuraṅga). Scholars tend to refer to this artistic style as the 'Bình Định Style' and date it to the 12th to 14th century, which is a period during which local forms of Hinduism appear to have increased in prominence (Lê and Nguyễn 1992; Ngô 2011, p. 289). All this evidence suggests that while Núi Bà could have included works that were from a 6th to 7th century period of production, as Parmentier's account suggests, there was also a tower here of a 12th–14th century style.

Another element of the Núi Bà relics is the presence of legendary sea creatures from South Asian mythology (*makara*). Ngô (2002, p. 10) has compared these to the well-known 10th century *makara* of Mỹ Sơn Temple 1A: a series of large canine teeth in the upper jaw and a single canine tooth in the lower jaw, round eyes opening under an eyebrow, ringed with a decorative curve, with the entire head encroached a large flame-shaped decorative element. However, the *makara* of Núi Bà is different: There is no outward bend but an inward twist; the canine in the upper jaw is curved like a tusk, while there are two canines—one in the upper jaw and one in the lower jaw—that are large and elongated; both lips curl, while there is a short beard on the chin, the skull seems to disappear backward, and one ear is missing. This also indicates that the *makara* is more like the 12th century Tháp Mắm style associated with Bình Định province (Vijaya) (BTTPY 2003; Ngô 2002). Furthermore, the thin cloth that is around the center of several of the bodies and the *mukuṭa* headdress that

Parmentier described are also features of the Tháp Mắm style. Ngô (2011, pp. 290–92) has also dated one inscription in the area to 1333 *śaka* or 1411 CE. All this evidence suggests that these constructions could be associated with an era of construction in Phú Yên where the locality was more subordinate to the influence of the Vijaya polity further northward.

In July 1999, in Mỹ Thạnh Tây village (nearby Núi Bà and Phước Tịnh), locals stumbled on a special find, which Ngô (2011) suggested was a Buddhist relic. The upper part of the stone object had been broken, although the object was egg-shaped and sandstone (0.97 m high, 0.65 m wide, and 0.1 m thick). On one side of the object was a Buddha image, seated, with hands poised in a meditation *mudrā*. Curved ridges along the outline suggest a sun-disk (*prabhāmaṇḍala*) behind the head of the Buddha, and there is a bump on the Buddha's head (*uṣṇīṣa*). On either side of the Buddha, there are symmetrical temple images, with square bases, round tops, and a parasol-like roof structure, with 10 canopies and a circle at the very top. The lower part of the slab was in the shape of a blooming lotus. The dating of this Buddha image is not clear, as with many finds of the area (Lê and Nguyễn 1992; Ngô 2011, pp. 54–61). However, this is where historical analysis can begin to help us unravel some of the mysteries of the area.

### 4. Historical Analysis of Local Champā Context and Transregional Connections

The archaeological evidence of the Hồ Citadel and surrounding Champā sites points to periods of settlement dating to the 2nd–5th centuries, the 6th–7th centuries, and the 12th–15th centuries. As the Hồ Citadel was situated at the banks of the Đà Rằng River, the largest and only river that stretches deep into the hinterlands at the foothills of the Annamite Chain and up into the cordillera, it is a prime location as a point of access into the uplands. Upstream–downstream trade networks would have benefitted greatly from access to hinterland products, especially eaglewood, which is also known as aloeswood, and it is named *gharu* in Portuguese and *gihlau* in Cham. Eaglewood has retained great symbolic importance in contemporary Cham culture, ritualistically symbolizing trade connections between uplands and lowlands, the power of monarchs to control luxury items, and also connecting human and divine realms (Ken 2011; Noseworthy 2013, 2015). To the west of the Hồ Citadel is the newly discovered Đồng Miếu site, while Núi Bà is to the south. To the east is the relatively flat plain of the Đà Rằng River valley. Further to the west of the Hồ Citadel, we know of many Champā relics that have been found, including the terracotta statues of Củng Sơn, the inscription found Tư Lương (Tân An, Đak Pơ), or the remnant foundations of the Bang Keng tower in Krông Pa, as well as the remnant foundations of the Yang Mum and Drang Lai, towers in Ayun Pa, in addition to the remains of Kuai King at Ayun Pa in Gia Lai province. Therefore, it is probable that the Hồ Citadel was a 'gateway to the hinterlands' for Champā (Nguyễn 2010, pp. 26–28; Ngô 2001, pp. 55–60; 2011, p. 28; see Figure 5).

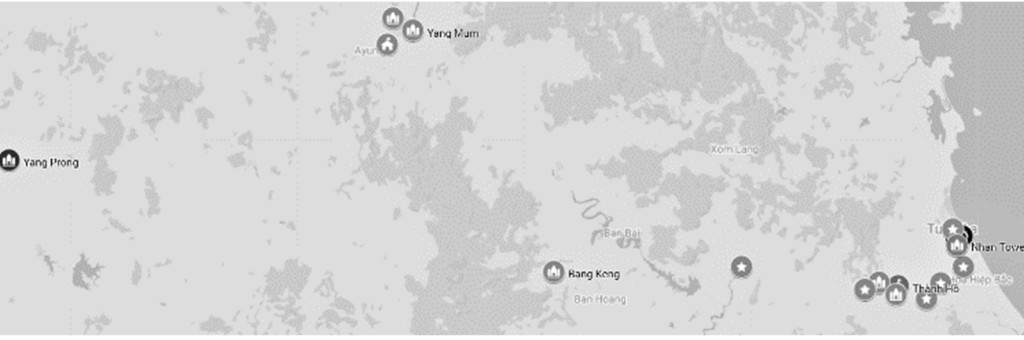

**Figure 5.** Approximate distribution of Champā sites in Phú Yên, Gia Lai, and Đắk Lắk provinces (Designed in Google Maps by Noseworthy).

Many of these hinterland Champā sites date to much later periods, as late as the 15th century. Still, the most recent archaeological analysis suggests very early settlement in the

region, especially in the vicinity of the Hồ Citadel ([Lâm 2019](#)). Chinese and Vietnamese historical records, however, might fill in some of the gaps of our understanding here. The consideration of historical records does more than fill in the gaps in our understanding, however, as the analysis of these sources also provides us with hypotheses to be investigated with future archaeological research.

The Tang dynasty prime minister Jia Dan compiled a series of itineraries around 800 CE as the *Huanghua sida Ji* (皇華四達記). Although these are now lost, they are quoted in the 'New History of the Tang' (*Xin Tángshu*, 新唐書), and past scholars believed the 'Mendu' (門毒) mentioned in these references was Vijaya. However, more recent scholarship tends to identify *Mendu* with Phú Yên ([Shiro 2011](#), p. 145; [Đỗ 2016](#); [Barocco et al. 2019](#)). [Shiro (2011](#), p. 130) proposes that, during the Song–Yuan period, *Longrong* (or Nongrong (弄容)) in the 13th century Song Dynasty work by Zhao Rukuo ('A Description of Barbarian Nations' (*Zhu Fan Chi*, 諸蕃志)) is a transcription of 'Ran Ran', which was mentioned later European accounts as a key port or river in what we now call Phú Yên. As is well recorded in local oral histories, when the Vietnamese Lê dynasty army moved through this area in the 15th century, they stopped at Thạch Bi, a stone marker just south of what was 'Răn Răn' and is now known by the Cham-Việt name: Núi Đá Bia. Whether local Vietnamese recognized 'Bia' as the Cham word for a female sovereign, often used for a goddess, is another question.

Looking back into the 16th century, the *Đại Nam Thực* Lục (大南寔錄, 1844–1909 CE) provides us with a record of the Hồ Citadel, which the geographical compilation calls An Nghiệp.

> ' . . . An Nghiệp Citadel at the north of the Đà Diễn River, belonging to An Nghiệp commune, Tuy Hòa district, has a circumference of about 5600 m. Legend has it that the Champā people built it and called it Hồ Citadel. In the Year of Mậu Dần (1578) of the Thái Tôn Dynasty, the Quận Công Lương Văn Chánh invaded this citadel, which has a long history'. ([ĐNTL 2007](#), p. 87)

The *Đai Nam Nhất Thống* Chí (大南一統志, 1882 CE) also includes a passage stating, 'Lương Văn Chánh from Tuy Hòa district, defeated [a] Champā [army] and was promoted to Superior administrator of Trần Biên. He had the merit to recruit and reclaim the fields. When he died, he became [a local] deity' ([ĐNNTC 1997](#), pp. 93–94). Indeed, there is a local understanding that Dinh Ông—also the name of a contemporary bridge in honor of the figure—is indeed dedicated to General Lương Văn Chánh after his death fighting against the Hồ Citadel in the 16th century. Dinh Ông Mountain is located to the west of the Hồ Citadel, which is close to Highway 25 (previously: Provincial Road 7). This is understood as being opposite of Núi Bà in Vietnamese understanding, which is on the other side of the Đà Rằng River. These finer points of evidence point toward Champā occupation and control of the Hồ Citadel during the 16th century. However, archaeological surveys have yet to take up this period as the focus of any research. Furthermore, the 16th century is even rarely mentioned in the more detailed historical studies that we have cited previously in this article. Instead, they tend to jump straight from the 15th century conquest of Vijaya to the 17th century conquest of the area associated with Phú Yên. What might we make of an account of this region from other sources?

Curiously, however, the early 17th century Ming-Chinese map known as *The Selden Map* uses a term that typically refers to Champā to describe what appears to be the Phú Yên-Khánh Hòa region, *Zhànchéng* (占城), along with another curious term, *Luówāntóu* (羅彎頭), to presumably refer to Pāṇḍuraṅga ([Bodleian Library 1620](#)). The map itself appears to be predominantly for the purpose of trade; given that the port of Aia Ru (i.e., Phú Yên province) was an important access point for valuable hinterland products, including, but not limited to, eaglewood (*gihlau*) and ivory, it does follow that Aia Ru could have been a point of interest. Furthermore, although the map is thought to date from the 1620s, it was not uncommon for Chinese maps to be slightly 'behind the times' with regard to the political changes in Southeast Asia. Thus, we can suspect that this confirms the idea that the area of Phú Yên, which Cham manuscript sources tend to refer to as Aia Ru, was indeed

controlled as a form of Kauṭhāra-proxy to Pāṇḍuraṅga in the early 17th century to the extent that it was important enough for late-Ming dynasty era Chinese map makers to record this as where, likely, the northern borderlands of Champā (Zhànchéng [占城]) were. The fact that Aia Ru (Phú Yên) and Aia Trang (Khánh Hòa) were still under Champā control in the early 17th century according to Cham language sources helps explain the passage in *Đại Nam Nhất Thống Chí* that describes events occurring in 1611 CE (Tân Hơi), as ' . . . the Champā troops invaded the border. The Emperor sent Lord Văn Phong [unknown to the locals] to take troops and capture [the land]. He established a palace for Đồng Xuân and Tuy Hòa districts to rely on Văn Phong, who was sent to save that land' (ĐNNTC 1997, p. 36; see also: Ken 2011, p. 243). The passage refers to Nguyễn Hoàng's expansion of his territorial control over several settlements, occupying lands from the Cù Mông Pass to the Đá Bia Mountain (Thạch Bi). He then regulated Tuy Hòa to Quảng Nam and Đồng Xuân to Phú Yên. From the 17th through the 20th century, it became common to reuse Champā materials in Vietnamese constructions. This includes the image of a three-headed, six-armed goddess, which is clearly adopted from the religions of Champā at Núi Bà Pagoda, and materials adopted into the construction of several Nguyễn era tombs. This helps explain why, in the late 18th century *Bình Nam Đồ* 'Map of the Pacified South', the panel labeling Phú Yên does not have any mention of evidence of Champā constructions (Figure 6).

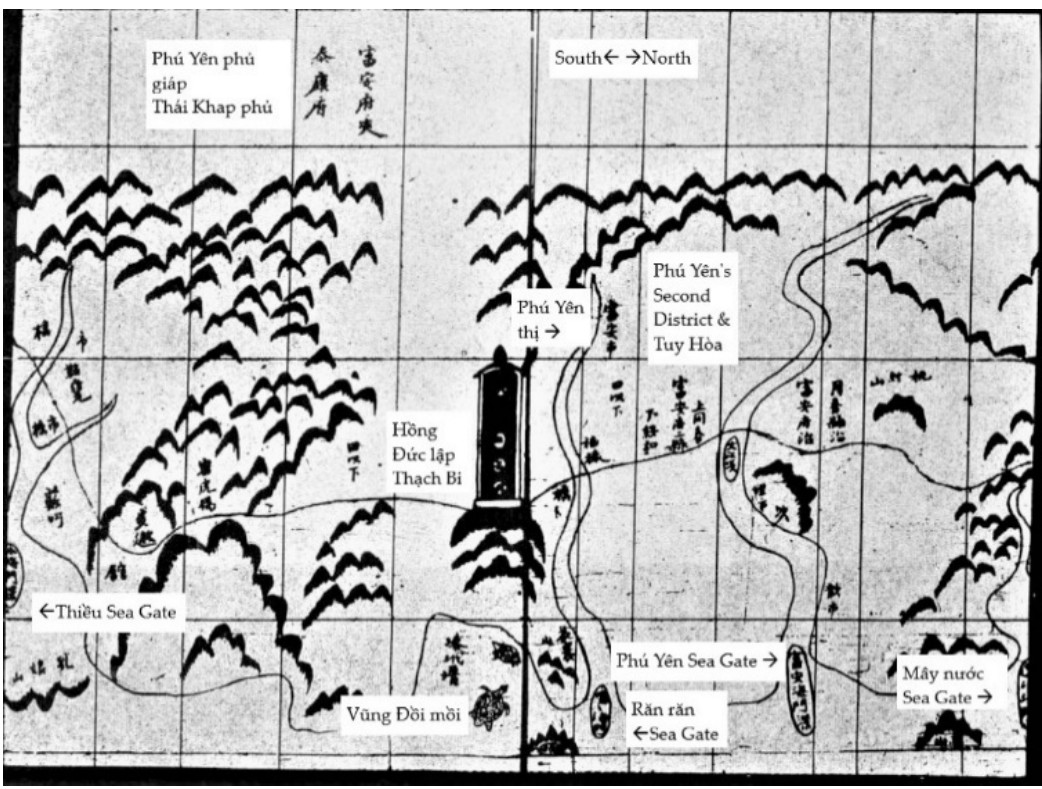

**Figure 6.** *Bình Nam Đồ* panel of Phú Yên province in classical Vietnamese, with romanized Vietnamese added to key locations (adapted from Bửu et al. 1962).

In the above panel of *Bình Nam Đồ*, we have evidence of Thạch Bi, which was symbolically important for the Nguyễn, by terms of connecting their legitimacy back to the previous Lê dynasty, even if that connection was more a matter of imagination than historical fact. The lack of evidence of the Hồ Citadel, Núi Bà temple, Nhạn Tower, and other Champā sites on the *Bình Nam Đồ* can be read very simply as what it was: erasure from the historical record as a result of conquest. This is not to say that an individual drew the citadel on the map but then erased it, or that they intentionally left it out, although the latter possibility is certainly more likely than the first. Rather, it suggests that the drafters of

the *Bình Nam Đồ* were not motivated to place a citadel, now in the minds of the conquerors aptly reduced to rubble, on their map. Hence, from 1611 CE to the establishment of the French Protectorate of Annam (1883 CE), this area was subject to the general decay of the sands of time, the shifting silts of the river, and the gradual appropriating of Champā relics by an ever-increasing Vietnamese population. Consequentially, the shift in religious community would be a decline in the influence of Hindu religion in the area, while the Buddhist and Hindu elements of Champā religion were incorporated into the local structures of Vietnamese and Sino-Vietnamese Buddhism. Furthermore, any brief Early Modern Muslim influence in the area does not appear in the records of the 19th century. Nonetheless, by the 19th and 20th centuries, interest in the study of Champā increased, and Hồ Citadel and the surrounding sites were given increased recognition. We can even still clearly see the outline of the old ramparts in contemporary satellite imagery, even as modern and contemporary residential and commercial constructions bump up against the archaeological footprint of this historic citadel (Figure 7).

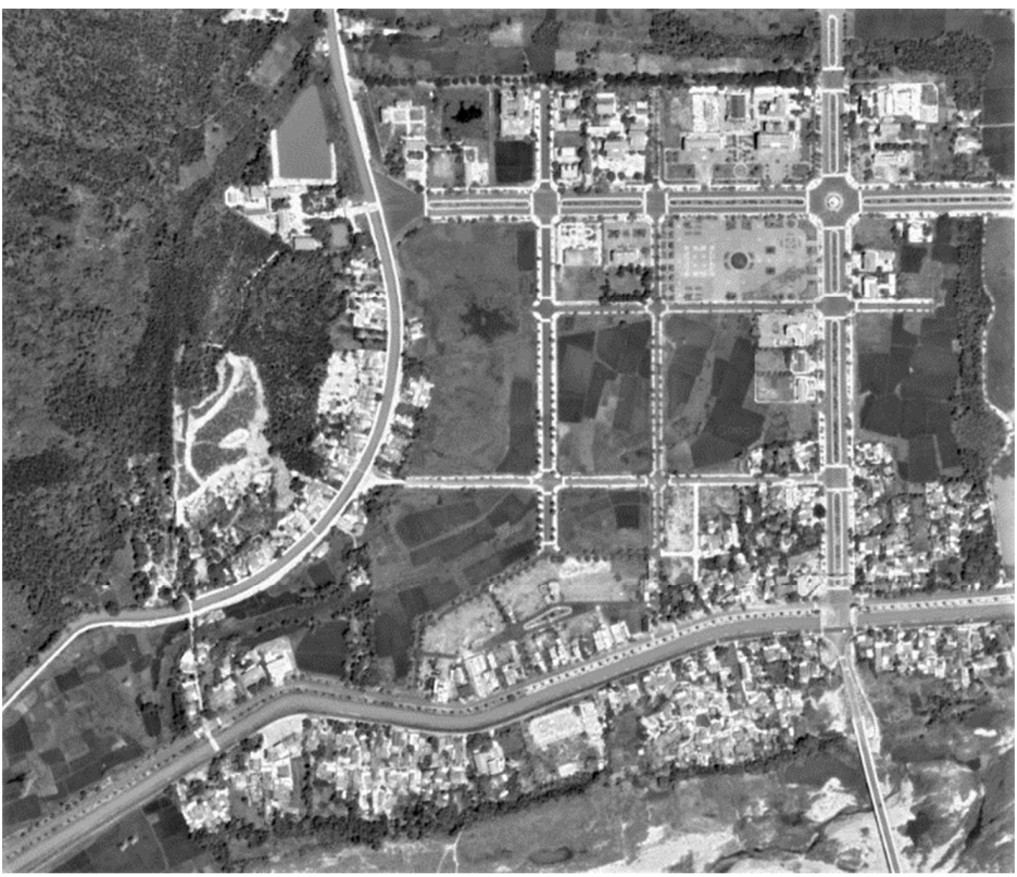

**Figure 7.** Satellite Image of Hồ Citadel (Adapted from Google Earth by Noseworthy).

In contemporary Vietnam, major thoroughfares cut through the notable Champā citadel. There are government offices for Phú Hòa district, a local ATM for Agribank, a district police station, and other such semi-official offices located within the area that was once enclosed by ramparts. Streams no longer traverse the area as definitively as they did during the early 20th century, but there appear to be significant portions of the interior that have been transformed into agricultural plots for local Vietnamese families. In the southern and western sections, significant settlements include residential households and *quán* eateries. While the contemporary Cham populations of provinces southward still call this area Aia Ru—comparing it to Aia Trang (Nha Trang) slightly further southward—the local creolized language of the Haroi ethnic minority refers to the Đà Rằng River as Ea Pa or Ia Pa. Haroi language became creolized precisely as a result of Champā peoples

who hid in the hinterlands to avoid being assimilated into the Nguyễn Vietnamese culture. While H'roi is typically classified as a Bahnaric (Austroasiatic) language by contemporary linguists, the geography and history of this area suggest strong Champā (Austronesian) ties and future historical research could investigate these links in greater detail.

## 5. Conclusions

Typically, historians and archaeologists of Champā have seen Phú Yên sites as part of the Kauṭhāra polity, centered at Nha Trang, Khánh Hòa province, and only refer to it as a piece of the Hindu–Buddhist Champā civilization that was incorporated into Vietnamese territory. However, we have shown that there is plenty of evidence to suggest that this area developed at first during the Lín Yì period (2nd–5th century) and continued to be a local polity as 'Kauṭhāra' (6th–9th century) before it fell out of prominence within the context of Champā. Regardless, with the 12th century emergence of the Vijaya polity centered at Quy Nhơn, Bình Đình province, the areas around the Hồ Citadel, including Núi Bà and the Nhạn Tower, were part of the Vijayan cultural sphere during a period when historians have suspected that the overwhelming Hindu–Buddhism court culture increasingly gave way to forms of localized Hinduism. Although the influence of Vijaya was destroyed with the conquest of Vijaya in 1471 CE, when the Lê dynasty army came all the way south, through this region, to Thạch Bi, they did not stay. Thus, the influence of Vietnamese Buddhism was not strong during the Early Modern period (15th–17th centuries). Lê dynasty military incursions would not equate to Lê dynasty control of the Đà Rằng River valley and the area returned to Champā control.

We relied upon historical documents to suggest that the area of Phú Yên retained an association with Kauṭhāra and/or Pāṇḍuraṅga, but probably primarily Pāṇḍuraṅga, from the 15th through the 17th century, until the period of Nguyễn-Vietnamese conquest. During this period the socio-religious dynamic of the community would have been increasingly a mix of localized Hinduism and Islam. However, a new period of Vietnamese settlement is marked, shortly thereafter, by the founding of the Bảo Tịnh and Hồ Sơn Buddhist temples in the 17th and 18th centuries, respectively. Throughout our analysis, we have observed transregional connections from Southeast to South Asia as an example of a 'Southern Asia' religious network. Yet the incorporation of sites into Vietnamese Buddhism also suggests that an 'Eastern Asian' religious and trading network was prominent as well. How these networks intersected with the networks of Muslim traders remains unclear based on the evidence we have at hand and should be considered in future research and discussions of potential new archaeological investigations in the vicinity. Indeed, we have also seen evidence of a transregional Southeast and East Asian *Nanhai* trading network through the inclusion of this area on *The Selden Map*.

Combined with Vietnamese sources and Cham sources, it is possible that further considerations of European and Chinese historical sources on this region could provide excellent leads for research in early modern archaeology. However, we also are obligated to emphasize a key finding of our fieldwork that is not yet mentioned in our historical considerations of Phú Yên: We found that many Cham and Haroi community members felt they were left out of discussions of 'what to do' with these heritage sites that were clearly related to the pasts of their communities. The future implications of our study are two-fold: First, that future studies ought to take into greater account the needs and concerns of Cham and Haroi communities over their heritage sites and any potential archaeological explorations beginning to look into the early modern period; second, that interdisciplinary studies of Champā sites yield fruitful results, especially when the *longue durée* history of a series of sites is considered, which could then be used to highlight gaps in the existing archaeological evidence and provide considerations for future archaeological studies. Thus, we hope to inspire continued and more direct collaborations between historians and archaeologists for the benefit of advancing research in the study of local and transregional understandings of Asia in social sciences.

Indeed, scholars have recently highlighted the need to develop integrated studies of Champā archaeological sites into a more cohesive understanding that includes the perspectives of Cham stakeholders and local provincial departments (Ngô et al. 2020; Quảng et al. 2020). Champā culture is a rich source of heritage for the indigenous Cham communities of Vietnam, although it also informs how scholars understand this area in the context of the *longue durée.*

**Author Contributions:** V.S.Q., who is an archaeologist and a member of the contemporary Cham community, visited Phú Yên first in 2009 on a survey and several times thereafter. He provided one photograph for this article, the conceptualization of the initial research questions, the archaeological data, and initial drafting of parts of the article relevant to his discipline. W.B.N., having also visited sites in the province, provided substantial theoretical framing, historical analysis, and wrote much of the article in conversation with V.S.Q. W.B.N. also designed the rest of the images, all the while in consultation with V.S.Q. All authors have read and agreed to the published version of the manuscript.

**Funding:** Funding was provided for V.S.Q.'s research by Van Lang University.

**Institutional Review Board Statement:** Not applicable.

**Informed Consent Statement:** Not applicable.

**Data Availability Statement:** Not applicable.

**Acknowledgments:** Beyond Văn Sơn Quảng, we send our gratitude to members of local Vietnamese and Cham communities who helped guide our research. This study has been supported by Van Lang University, McNeese State University, and Cornell University.

**Conflicts of Interest:** The authors declare no conflict of interest.

## Notes

[1] Historically known as the Sông Ba River.

[2] He argues the other is the Trà Khúc River valley of Quảng Ngãi Province.

[3] Scholars normally refer to this as 'Thanh Ho', or 'Thành Hồ' with proper diacritics. However, since 'thành' [城] is simply a Nôm (Demotic Classical Vietnamese) and Hán-Việt (Sino-Vietnamese) term for 'citadel', adopted from the Classical Chinese for 'city walls' and most often translated from Vietnamese to English as 'citadel', we refer to this location as the 'Ho Citadel of Champā' or simply 'Ho Citadel' throughout our article. It should not, however, be confused with the 15th century Ho Citadel of Thanh Hóa province, better known as the 'Citadel of the Hồ Dynasty'.

[4] There are many meanings of the term *negara*. Rather than entertaining the debate over whether the Euro-centric term "city-state," the antiquated French construction "principality," or the more literal "kingdom" is apt here, we simply gloss the term broadly as "polity," meaning an organized socio-political unity.

[5] A Vietnamese name derived from the Cham words: Aia Trang. Aia here means 'water' and 'Trang' refers to a place where hot and cold water mix, in this case, salt and fresh water.

[6] This tower is known as Yang Kơ Hmong in the Ede and Jarai languages, which are two Chamic Austronesian languages related to Cham. Both the Ede and the Jarai are considered 'peoples of Champā' who probably moved into the hinterlands to avoid incorporation and assimilation into Vietnamese society in the late classical period.

[7] For the Romanization of Cham language terms in this article we simply follow the American Library of Congress standard. This tower is also known as Tháp Bà in Vietnamese.

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
