# Peer review of "A View of Champā Sites in Phú Yên Province, Vietnam: Toward a Longue Durée of Socio-Religious Context"

_religions, doi:10.3390/rel13070656_

Round 1

Reviewer 1 Report

This is a very documented paper in terms of archaeological findings. Yet, it is too descriptive and lacks strong and convincing analysis. The structure of the article needs to be strengthened. The descriptions are detailed, yet there is a lack of analysis of the data provided. How do these archaeological findings and sites inform us about Champa history and its socio-religious evolution? Why are they important for our knowledge of the region? What do they tell us regarding the relationship between Champa principalities and between Champa and Vietnam? These are questions that are not addressed clearly in the paper. Furthermore, I am not convinced about the use of the term ‘longue durée’. Why is this term relevant to the present paper? The author should indicate

TITLE: The title needs to be rephrased. What do the authors mean by ‘local and transregional socio-cultural context’?

ABSTRACT:

The abstract is too long. It should feature the main points of the paper. It should also not contain any references.

KEYWORDS:

The list of keywords needs some work. I do not think that ‘interdisciplinary methods’ and ‘longue durée’ should be listed as keywords. I would rather like to see: archaeology; Central Vietnam; Champa religions etc.

Chapters 2 and 3 need thorough restructuring. First, the sites and the archaeological findings, then the religious iconography. The authors should indicate the significance of the sites and the findings both for our understanding of the history of Champa and the region as well as the religious evolution of this region.

There is relatively much information regarding the Ho citadel. However, why is this citadel important? How the study of the citadel gives us insights into the historical and religious history of Champa and the region?

Chapter 4 needs some work. It is mostly descriptive and not analytic. There is very little ‘historical analysis’. What should we understand from the sources mentioned by the authors? The authors should give us their own analysis of the sources rather than letting the reader draw their own conclusions. Furthermore, there is neither description nor analysis of the ‘transnational networks’. Furthermore, in which context should we understand ‘transnational networks’? Do the authors mean cultural and religious exchanges?

The study of maps is interesting. However, the authors do not provide enough information on their origin, their content, and their purpose. The production of maps is tied to specific purposes such as for instance military, diplomatic, or trade purposes. The Binh Nam Do is a strategic map used for war purposes. Furthermore, I do not understand why the authors chose to compare the Selden Map and the Binh Nam Do

The conclusion should be rewritten. It should summarize the main points of the paper and ask new questions. I have neither seen an analysis of ‘transregional connections’ nor an analysis of the region’s connections with India. Furthermore, the authors mention the incorporation of Champa religious elements into Vietnamese religion but that point is not adequately developed in the article. Finally, I do not understand what the authors mean by “Southern religion”.

Here are some points that need to be corrected and/or adressed:

Line 22-25: should be removed

Line 32: I would use “principality” instead of “kingdom”

Line 33-35: please provide the Cham/Sanskrit names of the principalities

Line 59-62: to be removed

Line 85-86: what is the use of this statement?

Line 89: “close” to what?

Line 92: what does “Vietnamese nationals” mean? Vietnamese scholars?

Line 99: what does “transnational connections” mean in this context?

Line 108: Please provide the names of the people who did the surveys

Line 101-125: what are the main hypotheses?

Line 127-128: the authors should explain the term ‘negara’. ‘Negara’ has a variety of meanings. What does that word mean in this context?

Line 178-180: I am not convinced by the argument. Furthermore, why is it relevant here?

Line 205-206: This explanation seems a bit simplistic. Hindu ascetics, yogi, or sadhu for instance do not wear any jewelry or clothes, and that does not make them Buddhist. The authors should also be mindful of not confusing ‘Brahmins’ with ‘Hindus’

Line 233-235: Lakshmi is the shakti of Vishnu. The term ‘consort’ is a Western understanding.

Line 329-330: why was gaharu important? What is its significance in Champa religions and culture? Any reference? Any other example of upland-lowland trade?

Line 382-383: sources? Reference?

Line 406: “erasure”? I do not understand why the authors use this term. There could be numerous reasons for the absence of mention of the Ho citadel. Was the citadel even visible at the time of the production of the Binh Nam Do map? If remains of the citadel were clearly visible, what would have been the purpose of erasing it from the map?

Line 413-414: I do not see the authors’ point. Why mention Muslim influence here?

Line 429-435: Any evidence? Any study on that topic?

Line 472-476: I do not see the relevance of this statement.

Author Response

Both reviewers make excellent points with regard to the improvement of our draft article. We have endeavored to carefully complete their suggested edits, as well as explain the reasoning behind the details on which we differed from the opinions of the reviewers, based on our closer understanding of the evidence at hand. The second reviewer mentions that all references cited are relevant to the research and the article is clearly referenced, and thus we did not endeavor to improve too much on these areas, but rather spent most of our efforts addressing the points under consideration by both: improving the context, explanation of methods, arguments, structure, clarity of presentation, and conclusion. We are mostly concerned with the text of the article for now. If the editors find the revisions sufficient that we have moved on to the stage of “minor revisions and copy editing” we will most certainly add improved images and additional maps as suggested by the reviewers.

The first reviewer makes excellent points for the revision of the title and abstract, along with the keywords. However, we disagree with some of their assertions. This is a coauthored article between a historian and an archaeologist. We do not think “archaeology” needs to be a keyword, much like we do not think history needs to be a keyword. Instead “interdisciplinary methods” is significant for us, but we want a middle ground between us and the reviewers. How about “historical archaeology.”?

Keywords operating as a part of the metadata of the article are designed by us to grasp the paper. Since one of us has specific training in how metadata operates (beyond normal scholarly training in metadata) we think are keywords are designed well to help interested readers find our paper.

We have endeavored to revise the paper with the following considerations: Sections 2 & 3 have been thoroughly restructured. We addressed the other points raised in the revision of the document.

Section 4 added quite a bit more analytical points, rather than descriptive points. We also motivated the comparison of the maps. We also addressed the other points raised here in the revision of the document, including the motivation of using the maps. We rewrote the conclusions.

The finer details we address 1 by 1:

Line 22-25. The reviewer does not explain why they think these should be removed. They are simple statements of fact that are important in the abstract of the paper, which assuredly a reader would get to before the introduction.

Line 32: The reviewer uses an antiquated French term here. We should explain here, but do not need to detail in the piece: we are not at all concerned with orthodox interpretations biased by Vietnamese-supremacist interpretations of this history that rely on a preference for Vietnamese sources. We think “kingdom” is a better term and we will stand by that usage. But, we also want to compromise since this is not the point of the article. How about the more neutral term “polity”? We also adjusted the language to use polity.

Line 33-35: Vickery’s point is not about the superimposition of Champa (not ‘Cham’; but ‘Champa’)/Sanskrit names on these regions. Others have done so, inaccurately assuming a 1:1 relationship between negara kingdoms and corresponding riverine regions. Since this is not the focus of our paper, we do not need to repeat these errors here.

Lines 59-62: we revised, but did not remove these lines.

Line 92: Vietnamese nationals includes people who are citizens or legal residents of Vietnam but not ethnic Vietnamese. This is the proper term here. Vietnamese scholars implies all are ethnic Vietnamese, which is incorrect. We used the correct term and so we do not need to change it.

Line 99: We revised this line considering the suggested scope of the article by both reviewers.

Line 108: The comment of the reviewer is unclear. These hypotheses are found in the combined citations of Le & Pham, Lam & Nguyen, and Lam. We cited these works already.

Line 101-125: The reviewer asks “what are the main hypotheses?” This section summarizes the three main hypotheses. They must have found the structure of the text confusing. Hence, we thoroughly revised this section to clarify.

Line 127-128: There are so many meanings of negara it is worth a separate article. This is a rabbit hole we should not go down at this time. We simply gloss the term as “polities” here and that is sufficient.

Line 178-180: We are summarizing previous archaeological finds in the area, while pointing out that by the early modern period the area had Champa sites that were incorporated into Vietnamese Buddhist practice. We clarified the wording.

Line 205-206: We are not confusing Brahmins with Hindus, although this explanation is probably in the reviewers mind because in French scholarship the term Brahmins is often used for Hindus. We mean brahmins. Regardless, we revised this transition.

Line 233-235: To describe “consort” as “Western” is also a mistake. It’s an English language gloss, but well accepted throughout the world. Furthermore, other “Western” languages actually have different glosses for the concept of shakti. The French “puissance” for example, is also a “Western” gloss. French, German, English, they all have their biased views about how to translate shakti. Since we are writing for an English language readership, we needn’t go to deep here.

Line 329-330: We provided an explanation here.

Line 382-383: The reviewer misread this quote. There was already a reference. We made it a block quote to make it more clear. Or did they mean the previous line about Thạch Bi? This is common knowledge in the area. We clarified it is part of the oral historical record.

Line 406: The reader is reading the term erasure almost as if a person drew the citadel on the map and then erased it. This is not what erasure means in this usage. There are multiple meanings of the term. Historical erasure after conquest means that the Vietnamese source material was not motivated to map out a site their armies.

Once again, we are extremely grateful for the efforts of the reviewers and editors of this text. We hope that they will find the piece is sufficiently improved to move on to the stage of “minor revisions and add suggested images” but want to underscore our commitment to move this paper through the stages of publication as the editorial board sees fit. Should such revisions not be required, we would be delighted by that result as well.

Reviewer 2 Report

From what I understood of this paper, you take issue with the fact that previous scholarship on the Cham, particularly the Ho Citadel site, has been short-sighted - choosing to focus on a single period. What you as authors have contributed is an expanded perspective on the long history of the site. You have done this by reviewing archaeological data - in general in section 2 (63-125) and in great detail in the discussion of objects discovered by Parmentier in section 3.2 (187-255) - and textual data and maps - in section 4. 

Moving forward, I think you need to identify a) a main argument and b) what the main contribution is (select one or more of the proposed conclusions highlighted in the final paragraph) and rewrite the paper to ensure that you truly accomplish these (supporting/proving the argument, and emphasizing your academic contribution). Better written organization (e.g. topic sentences at the beginning of paragraphs) would also make the paper easier to read.

Having read the paper multiple times over I think the significant academic contribution is the bridging of various historical narratives into one long narrative. If you feel strongly that your project aims to conceptualize the Cham civilization through the lens of longue duree then that really needs to be put in the beginning and fleshed out in much greater detail: what longue duree is, what it is not, what theoretical and conceptual work it is doing in your historical analysis, why this is not only different than previous approaches but better or a new significant contribution to the studies of this civilization (or perhaps just the site or region). I would also be interested to hear how you see the different sources working together - rather than "filling gaps" where do they overlap? How are they different? What is the benefit of drawing on both? Why is this important?

Author Response

(The authors gave the same response as above.)

Round 2

Reviewer 1 Report

The manuscript has been sufficiently improved to be published. The authors have made significant and careful revisions. There is no doubt that this paper will contribute significantly to our knowledge of ancient Champa and Vietnam.

Reviewer 2 Report

Overall, the authors have made noticeable improvements to the article, and have addressed many of the areas in need of reconsideration. As a result, the article is much stronger and I fully endorse it being published in this new finished state.